# Extrapulmonary tuberculosis in Pakistan- A nation-wide multicenter retrospective study

**Sabira Tahseen**[1,2]*, **Faisal Masood Khanzada**[1], **Aurangzaib Quadir Baloch**[3‡], **Qasim Abbas**[4‡], **Mansoor Manzoor Bhutto**[5¤‡], **Ahmad Wali Alizai**[6‡], **Shah Zaman**[7‡], **Zahida Qasim**[8,9‡], **Muhammad Najeeb Durrani**[10‡], **M. Khalid Farough**[11‡], **Atiqa Ambreen**[12‡], **Nauman Safdar**[13‡], **Tehmina Mustafa**[2,14]

**1** National TB Reference Laboratory, National TB Control Program, Islamabad, Pakistan, **2** Centre for International Health, Department of Global Public Health and Primary Care, University of Bergen, Bergen, Norway, **3** National TB Control Program, Ministry of National Health Services Regulation and Coordination Islamabad, Islamabad, Pakistan, **4** Department of Health, Provincial TB Control Program, Government of Khyber Pakhtunkhwa, Peshawar, Pakistan, **5** Provincial Reference Laboratory, Provincial TB Control Program, Sindh, Karachi Pakistan, **6** Department of Health, Provincial TB Control Program, Government of Balochistan, Quetta, Pakistan, **7** Department of Health, Provincial TB Control Program, Government of Gilgit Baltistan, Gilgit, Pakistan, **8** Department of Pathology, District Head Quarter Hospital Mirpur, Azad Jammu Kashmir, Pakistan, **9** Department of Health Azad Jammu Kashmir, District Head Quarter Hospital Mirpur, Azad Jammu Kashmir, Pakistan, **10** Department of Health, Ministry of National Health Services Regulation and Coordination, Islamabad, Pakistan, **11** TB Project, Green Star Marketing, Karachi, Pakistan, **12** Department of Microbiology, Gulab Devi Hospital, Lahore, Pakistan, **13** Global Health Directorate, Indus Health Network, Karachi, Pakistan, **14** Department of Thoracic Medicine, Haukeland University Hospital, Bergen, Norway

☯ These authors contributed equally to this work.
¤ Current address: Sindh Health Care Commission, Karachi, Pakistan
‡ These authors also contributed equally to this work.
* sabira.tahseen@gmail.com

**Data Availability Statement:** All relevant data are within the paper and its Supporting Information files.

## Abstract

### Background

Pakistan is fifth among high burden countries for tuberculosis. A steady increase is seen in extrapulmonary tuberculosis (EPTB), which now accounts for 20% of all notified TB cases. There is very limited information on the epidemiology of EPTB. This study was performed with the aim to describe the demographic characteristics, clinical manifestations and treatment outcomes of EPTB patients in Pakistan.

### Method

We performed descriptive analysis on routinely collected data for cohorts of TB patients registered nationwide in 2016 at health facilities selected using stratified convenient sampling.

### Findings

Altogether 54092 TB including 15790 (29.2%) EPTB cases were registered in 2016 at 50 study sites. The median age was 24 years for EPTB and 30 years for PTB patients. The crude prevalence of EPTB in females was 30.5% (95%CI; 29.9–31.0) compared to 27.9% (95%CI; 27.3–28.4) in males. The likelihood of having EPTB (OR), was 1.1 times greater for females, 2.0 times for children, and 3.3 times for residents of provinces in the North-West.

**Funding:** The author(s) received no specific funding for this work.

**Competing interests:** The authors have declared that no competing interests exist.

The most common forms of EPTB were pleural (29.6%), lymphatic (22.7%) and abdominal TB (21.0%). Pleural TB was the most common clinical manifestation in adults (34.2%) and abdominal TB in children (38.4%). An increase in the prevalence of pleural and osteoarticular and decline in lymphatic and abdominal TB was observed with advancing age. Diversity in demography and clinical manifestations were noted between provinces. The treatment success rate for all type EPTB was significantly high compared to bacteriology confirmed PTB with the exception of EPTB affecting CNS with a high mortality rate.

## Conclusions

The study provides an insight into demography, clinical manifestations and treatment outcomes of EPTB. Further studies are needed to explain significant diversities observed between provinces, specific risk factors and challenges concerning EPTB management.

## Introduction

Extrapulmonary tuberculosis (EPTB) represents 15% of the incident TB cases notified globally. The proportion of patients who present with extrapulmonary manifestations varies from 8% in the WHO Western Pacific region, 17% in South East Asia and 24% in the Eastern Mediterranean Region [1, 2]. There are many forms of extrapulmonary manifestations of TB, affecting every organ system in the body. Considerable differences are reported in susceptibility to different sites of EPTB by age, race/ethnicity, sex, and country of origin [3, 4, 5]. The proportion of EPTB among cohort of notified TB cases is reported and monitored every quarter at the country level and by WHO annually for all member countries, regions and global trends. Cohort of notified incident TB cases is analyzed for demographic characteristics collectively for PTB cases and for treatment outcomes, stratified by bacteriology confirmed and clinically diagnosed TB cases only. However, due to limited data variables used for routine quarterly TB surveillance reports, stratified cohort analysis for EPTB is not possible [6]. Routine surveillance data is used by a few, mostly high-income countries to study the epidemiology of EPTB including demography, clinical manifestations, bacteriological diagnosis and notification trends [3–5, 7, 8].

Pakistan is among the top five high burden countries (HBC). Implementation of the DOTS strategy for TB started in 2001 and within five years expanded to cover most of the public health sector. National TB control program (NTP) subsequently focused on the expansion of DOTs coverage in the private health sector, childhood TB and programmatic management of drug-resistant TB [9, 10]. Regardless, the management of EPTB remained mostly neglected, a steady increase is seen in absolute numbers of EPTB and proportion among notified TB cases from 17.4% (45,537) in 2011 to 20% (71,322) in 2016 [1,9]. Diversity is noted in the proportion of EPTB between provinces. Besides a few hospital-based studies, there is very little information on the epidemiology of EPTB disease in Pakistan [11, 12]. We performed a descriptive analysis on a nationwide sample to determine the demographic characteristics, clinical manifestations and treatment outcomes of EPTB patients in Pakistan.

## Study design and methodology

### Study design

This is a multicenter retrospective observational study, performed on routinely collected data for cohorts of TB patients registered nationwide in 2016 at health facilities selected using stratified convenient sampling.

## Study setting

Pakistan is a country in South Asia, with an estimated incidence of 510K TB cases and having a low prevalence of HIV. In 2016, 68% of the estimated TB cases were notified and 20% of these presented with EPTB [1]. The private health sector including general practitioners (GPs) engaged in TB care, contributed to 27% of all notified TB cases. By 2016, for TB diagnosis, microscopy services were established at about 1300 health care facilities (HCF), GeneXpert (Cepheid, Sunnyvale, CA, USA) at 73 HCF and TB culture in 16 laboratories [1, 10]. Xpert MTB/RIF (Xpert) testing is recommended for the diagnosis of EPTB since 2013 [1, 13] but TB culture facilities are offered mostly for drug-resistant TB patients. Histopathology services although limited to tertiary hospitals in public sector but are offered widely by commercial clinical laboratories.

Standard recording and reporting tools are used [6] and data of each notified TB patient is recorded in TB registers at each HCF. A separate TB register is maintained at district level for recording TB cases notified by GPs. Although there are no specific columns in TB registers for recording EPTB disease site or laboratory test results other than bacteriology but the staff is generally guided to record these details in a column for remarks.

Pakistan is administratively divided into four provinces; Punjab (PJB), Sindh (SND), Khyber Pakhtunkhwa (KP) and Balochistan (BTN), three regions namely Federally administered tribal area (FATA), Gilgit Baltistan (GB), and Azad Jammu Kashmir (AJK) and Islamabad capital territory (ICT). FATA districts were merged into KP province in 2019. SND and PJB are large provinces comprising >70% of the total country population and geographically located in South-East, whereas KP, FATA, BTN, and GB are in North-West of the country (S1 Fig).

## Data source and collection

We used TB registers with records of TB patients notified in 2016 for the study. For data collection, HCFs were selected using stratified convenient sampling. Separate lists were first obtained of HCFs in each province and region along with the numbers of PTB and EPTB cases notified by each in 2016. HCFs were than stratified into Level-I (Primary HCF/clinics), Level-II (secondary HCF/specialized TB hospitals) and Level-III (tertiary hospitals). Based on EPTB case notification, a shortlist was prepared by selecting the top five HCF/ tier for each province and three/ tier for the region. This list was then handed over to the respective TB program for data collection with guidance to further select from this list two HCFs/tier/province and one HCF/tier/ region, based on the quality and completeness of the data recorded in the TB registers. Data collection was started in 2018 and all HCF staff was guided to check for data completeness and to record missing data before making copies of the TB registers.

For a sampling of TB cases notified by private GPs, the two implementing partners working respectively in 13 and 79 districts were asked to provide a copy of the district TB registers. Private GP clinics were classified as level-I HCF.

## Data management

TB registers were received at NTP office Islamabad. A specially designed electronic file, (developed in EpiData Manager V4.4.2.1) was used for data entry. Case-based data was entered (through EpiData Entry Client v4.4.3.1, EpiData Association, Odense, Denmark) for variables including age, sex, history of TB treatment, disease site, laboratory results and treatment outcomes. All TB cases registered between 1st January and 31st December 2016 were included in the study. After entry, data was checked for completeness and if for any HCF, EPTB disease sites recorded were less than 80% of the registered cases, respective TB program was requested again to collect missing information where possible.

Standard definitions were used for defining children (0-14yrs), adult (≥15yrs), new, previously treated, bacteriology confirmed (B+) and clinically diagnosed TB cases based on data recorded in the TB registers [6]. Cases were categorized by major disease sites, reported as either PTB or EPTB. The PTB group comprised cases with PTB listed as the only disease site. The EPTB cases were grouped according to extrapulmonary disease site: pleural, lymphatic intra-thoracic, lymphatic extra-thoracic, osteoarticular spine and osteoarticular other than the spine, the central nervous system including meninges, abdominal including peritoneal and disseminated TB including miliary. Extrapulmonary manifestations if specified other than sites mentioned above were listed as "others" and if not specified were listed as EPTB not specified (NOS).

Data were analyzed using STATA® v13.1 (StataCorp, 4905 Lakeway Drive, College Station, Texas 77845, USA). Logistic regression was performed to calculate odds ratios (OR) and 95% confidence intervals for comparisons between groups, while mean, median and quartiles were analyzed for quantitative variables. Two sample proportion test was used to analyze differences in proportions between groups. A p-value of $p < 0.05$ was considered statistically significant.

Ethics statement: For confidentiality, patient identifiers were not entered into the database used for the study. The study protocol was approved by the "Institution review board" of HIV, TB and malaria programs, Islamabad, Pakistan and Regional Committee for Medical and Health Research Ethics, Western-Norway (REK Vest).

## Results

Altogether 50 TB registers were collected including 37 HCF (35 public and 2 private) and 13 district TB registers. Only one implementing partner provided records of TB patients notified by GPs in 13 districts. The study population covered 32 districts and included samples from all four provinces, three regions, and federal capital. Study sites included 29 level-I (16 public, 13 district GP clusters), 11 level-II (all public) and 10 level-III HCF (8 public, 2 private). Details of study sites are shown in S1 and S2 Tables.

### Reported tuberculosis cases

Altogether 54,092 TB including 38,302 PTB and 15,790 EPTB (29.2%) patients were registered at selected 50 study sites in 2016. Of all TB cases notified country wide in 2016, the proportion included in the study sample from each province/region is shown in S2 Table. Twenty one TB cases were recorded having concurrent EPTB and PTB and these were counted with EPTB cohort. Age was missing for 22 TB including three EPTB patients. The disease site was not specified for 1399 EPTB cases (8.5%). Treatment outcome was not recorded for 3222 TB (7.0%) including 694 EPTB (4.9%) patients.

The characteristics of PTB and EPTB patients included in the study are shown in Table 1.

### Demographic characteristics

The median age of patients with EPTB was 24yrs compared to 30yrs for PTB. Overall 52.4% of EPTB and 49.2% of PTB patients were females. Peak TB notification was seen in the age group of 15-24yrs (26.6%) with females being in the clear majority among both PTB (F: M 1.4:1) and EPTB patients (F:M 1.4:1). A decline in female TB notifications was observed with increasing age (Fig 1).

The crude prevalence of EPTB in females was 30.5% (95%CI; 29.9–31.0) compared to 27.9% (95%CI; 27.3–28.4) in males. Compared to the total study sample, the proportion of EPTB was higher among TB patients notified in North-West provinces/regions (BTN, KP, FATA, AJK) and ICT and are shown in Table 2 and S2 Table.

**Table 1. Characteristics of patients having extrapulmonary compared to pulmonary tuberculosis notified by study sites in 2016.**

|  | Pulmonary TB n(%) | Extra pulmonary PTB n(%) | OR (95% CI) |
|---|---|---|---|
| **TB cases notified** | **38302** | **15790** |  |
| Median Age (IQR) | 30(19,50) | 24(15,39) |  |
| **Sex** |  |  |  |
| Male | 19455(50.8) | 7519(47.6) | Ref |
| Female | 18847(49.2) | 8271(52.4) | 1.1 (1.1–1.2) |
| **Age Group\*** |  |  |  |
| 0–14 | 4731(12.4) | 3607(22.8) | 1.69(1.60–1.79) |
| 15–24 | 9932(25.9) | 4474(28.3) | **Ref** |
| 25–34 | 6474(16.9) | 2874(18.2) | 0.99 (0.93–1.04) |
| 35–44 | 4720(12.3) | 1649(10.4) | 0.78 (0.73–0.83) |
| 45–54 | 4946((12.9) | 1351(8.6) | 0.61 (0.57–0.65) |
| 55–64 | 4075(10.6) | 958(6.1) | 0.52 (0.48–0.56) |
| 65+ | 3405(8.9) | 874(5.5) | 0.57 (0.52–0.62) |
| **Place of origin** |  |  |  |
| PJB | 17950(46.9) | 5580(35.3) | Ref |
| SND | 12294(32.1) | 3514(22.3) | 0.9 (0.9–1.0) |
| KP | 3872(10.1) | 3930(24.9) | 3.3 (3.1–3.4) |
| BTN | 1310(3.4) | 801(5.1) | 2.0 (1.8–2.1) |
| FATA | 466(1.2) | 472(3.0) | 3.3 (2.9–3.7) |
| GB | 1256(3.3) | 437(2.8) | 1.1 (1.0–1.2) |
| AJK | 577(1.5) | 347(2.2) | 1.9 (1.7–2.2) |
| ICT | 577(1.5) | 709(4.5) | 4.0 (3.5–4.4) |
| **Previous TB treatment** |  |  |  |
| No | 34235(89.4) | 15072(95.4) | Ref |
| Yes | 3894(10.2) | 627(4.0) | 0.4(0.3–0.4) |
| NA | 173(0.5) | 91(0.6) | 1.2(1.0–1.5) |
| **Health Care Provider** |  |  |  |
| Public | 16684(43.6) | 9206(58.3) | Ref |
| Private | 21618(56.4) | 6584(41.7) | 0.5(0.5–0.6) |
| **Health Care facility Level** |  |  |  |
| PHC(All) | 16800(43.6) | 6128(38.8) | Ref |
| SHC | 9950(26.0) | 4687(29.7) | 1.3 (1.2–1.4) |
| TCH | 11552(30.2) | 4975(31.5) | 1.2 (1.1–1.3) |

PJB-Punjab, SND-Sindh, KP-Khyber Pakhtunkhwa, BTN-Balochistan, AJK-Azad Jammu Kashmir, GB-Gilgit Baltistan, FATA-federally administered tribal areas, ICT-Islamabad Capital territories. PHC-Primary health care, SHC-secondary health care facility, TCH-Tertiary care hospital

\* 22 cases with missing records for age excluded from this analysis

The odds of patients presenting with EPTB, compared with PTB, were 1.1 times as high for females, 2.0 times for children, 3.0 times for residents of KP and FATA and 2.0 times for BTN and ICT compared to PJB. Female to male ratio (FMR) and age-specific female likelihood to present with EPTB in each province /region is shown in Table 2 and S3 Table. A higher odds (OR = 1.3) for EPTB were reported among females in the age group of 25-34yrs and among residents of SND province (Fig 1 and S3 Table). No differences were observed in the likelihood of female presenting with EPTB seeking health care from the public (OR = 1.14, 95% CI: 1.08–1.19) compared to private sector (OR = 1.18, 95% CI: 1.11–1.24). Overall, 15.4% (8338) of all

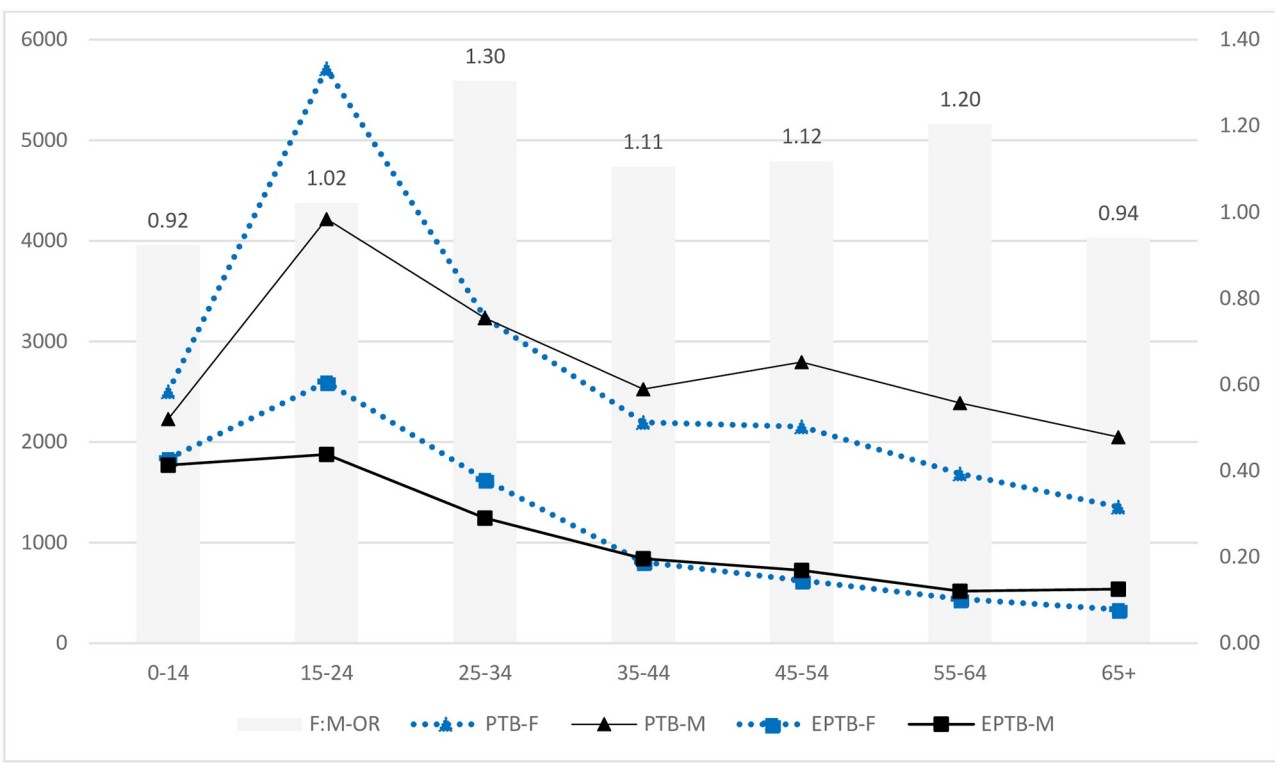

**Fig 1. Age and sex-specific pulmonary and extrapulmonary tuberculosis case notifications and age-specific odds of female for having extrapulmonary tuberculosis.** PTB-M; Male pulmonary tuberculosis patients, PTB-M- Female pulmonary tuberculosis patients, EPTB-F, Female extrapulmonary tuberculosis patients, EPTB-M, Male extrapulmonary tuberculosis patients, F: M-OR: Female to male odds ratio for EPTB. TB cases notified are shown on the primary axis and OR on the secondary axis.

TB, 22.8% (3607) of EPTB and 12.4% (4731) of PTB patient were children (Table 1). More than 40% of EPTB patients in BTN, KP, FATA, and GB were children compared to around 10% in PJB, SND, and AJK. The crude prevalence of EPTB in children was 43.3% and was highest compared to all other age groups. A decline in the proportion of EPTB was observed with increasing age (Table 3). A similar pattern was observed across all province/regions with the exception of SND having the highest proportion of EPTB among 15-34yrs (S4A–S4H Table).

### Extrapulmonary manifestations

The median age was youngest (17yrs) for patients with abdominal TB and oldest (34yrs) for patient with osteoarticular TB other than spine. The site-specific median age for the cohort of EPTB patients studied is shown in Table 3 and S4A–S4H Table.

Among all EPTB patients (15790), frequency of different manifestations were, pleural (29.6%), lymphatic (22.6%), abdominal (21.0%), osteoarticular (9.4%), central nervous system (CNS) (4.6%), other (3.7%) and disseminated TB (0.6%). The EPTB disease site was not recorded in 8.5% of patients. Frequency of different EPTB disease forms in children (3607) compared to adults (12180) were, abdominal (38.4% vs 15.8%), lymphatic (22.4% vs 22.7%), pleural (14.0% vs 34.2%), osteoarticular (3.9% vs 11.0%), CNS (6.0% vs 4.2%) and not specified (12.1 vs 7.4%) (Table 3).

**Table 2. Pulmonary and extrapulmonary tuberculosis case notified, female to male ratio and odds of female for having extrapulmonary TB.**

| | | Pakistan | PJB | SND | KP | BTN | FATA | GB | AJK | ICT |
|---|---|---|---|---|---|---|---|---|---|---|
| **All form TB** | All | **54092** | **23530** | **15808** | **7802** | **2111** | **938** | **1693** | **924** | **1286** |
| | Female | 27118 (50.1%) | 11759(50.0%) | 7746(49.0%) | 4026(51.6%) | 1149(54.4%) | 440(46.9%) | 952(56.2%) | 466(50.4%) | 580(45.1%) |
| | Male | 26974(49.9%) | 11771(50.0%) | 8062(51.0%) | 3776(48.4%) | 962(45.6%) | 498(53.1%) | 741(43.8%) | 458(49.6%) | 706(54.9%) |
| | FMR | 1.0 | 1.0 | 1.0 | 1.1 | 1.2 | 0.9 | 1.3 | 1.0 | 0.8 |
| **PTB** | All | **38302** | **17950** | **12294** | **3872** | **1310** | **466** | **1256** | **577** | **577** |
| | Female | 18847(49.2%) | 8819(49.1%) | 5763(46.9%) | 2044(52.8%) | 734(56.0%) | 234(50.2%) | 707(56.3%) | 302(52.3%) | 244(42.3%) |
| | Male | 19455(50.8%) | 9131(50.9%) | 6531(53.1%) | 1828(47.2%) | 576(44.0%) | 232(49.8%) | 549(43.7%) | 275(47.7%) | 333(57.7%) |
| | FMR | 1.0 | 1.0 | 0.9 | 1.1 | 1.3 | 1.0 | 1.3 | 1.1 | 0.7 |
| **EPTB** | ALL | **15790** | **5580** | **3514** | **3930** | **801** | **472** | **437** | **347** | **709** |
| | Female | 8271(52.4%) | 2940(52.7%) | 1983(56.4%) | 1982(50.4%) | 415(51.8%) | 206(43.6%) | 245(56.1%) | 164(47.3%) | 336(47.4%) |
| | Male | 7519(47.6%) | 2640(47.3%) | 1531(43.6%) | 1948(49.6%) | 386(48.2%) | 266(56.4%) | 192(43.9%) | 183(52.7%) | 373(52.6%) |
| | FMR | 1.1 | 1.1 | 1.3 | 1.0 | 1.1 | 0.8 | 1.3 | 0.9 | 0.9 |
| **EPTB% (95%CI)** | ALL | 29.2 (28.8–29.6) | 23.7 (23.2–24.3) | 22.2 (21.6–22.9) | 50.4 (49.3–51.5) | 37.9 (35.9–40.1) | 50.3 (47.1–53.6) | 25.8 (23.7–28.0) | 37.6 (34.4–40.8) | 55.1 (52.4–57.9) |
| | Female | 30.5 (30.0–31.1) | 25.0 (24.2–25.8) | 25.6 (24.6–26.6) | 49.2 (47.7–50.8) | 36.1 (33.3–39.0) | 46.8 (42.1–51.6) | 25.7 (23.0–28.6) | 35.2 (30.9–39.7) | 57.9 (53.8–62.0) |
| | Male | 27.9 (27.3–28.4) | 22.4 (21.7–23.2) | 19.0 (18.1–19.9) | 51.6 (50.0–53.2) | 40.1 (37.0–43.3) | 53.4 (48.9–57.9) | 25.9 (22.8–29.2) | 40.0 (35.4–44.6) | 52.8 (49.1–56.6) |
| **FM OR for EPTB** | | 1.14 (1.09–1.18) | 1.15 (1.08–1.22) | 1.47 (1.36–1.58) | 0.91 (0.83–0.99) | 0.84 (0.71–1.01) | 0.77 (0.59–0.99) | 0.99 (0.79–1.23) | 0.82 (0.62–1.07) | 1.22 (0.98–1.53) |

PJB-Punjab, SND-Sindh, KP-Khyber Pakhtunkhwa, BTN-Balochistan, AJK- Azad Jammu Kashmir, GB-Gilgit Baltistan, FATA-Federally administered tribal areas, ICT-Islamabad Capital territories, FMR-Female to male ratio, PTB-Pulmonary tuberculosis EPTB-extrapulmonary tuberculosis, FM OR-Female to male odds ratio

Among all ages, notification of pleural TB compared to abdominal TB was significantly higher ($P < .001$) in South Eastern provinces, PJB (36.2 vs12.0%), SND (33.9 vs 17.1%) and AJK (39.2 vs 16.4%) whereas, abdominal TB was more common ($P < .001$) in North West provinces including KP (34.4 vs 23.4%), FATA (49.8 vs 22.0) and GB (46 vs10.1%) (Table 3 and S4A–S4H Table).

Among adult population, pleural TB was the commonest form (range 16.9–55.6%) across all provinces/regions. However the second commonest was lymphatic TB in PJB, SND, GB and ICT (21.3–26.7%), compared to abdominal TB in KP, BTN and FATA (range 15.3–25.3%). Among children the most commonly reported extrapulmonary manifestation was lymphatic TB in PJB, SND and AJK (>40%), abdominal TB in KP (44%), FATA and GB (>70%) and meningitis (33%) in BTN (S4A–S4H Table).

A steady decline in lymphatic and abdominal TB and an increase in pleural and osteoarticular TB was observed with advancing age (Table 3 and Fig 2).

Lymphatic TB was significantly more common in females (26.6 vs 18.4%, $P < .001$). A higher number of females were observed within each EPTB disease form, with the exception of pleural TB which was more common in males (34.5 vs 25.1%, $P < .001$) (Table 3).

## Bacteriological diagnosis

Results of AFB smear and/or Xpert were recorded for 86.1% PTB and 8.8% EPTB patients. Among notified cases, 49.3% of PTB (18872) and 0.55% (87) EPTB patients were bacteriology confirmed. Culture and/or histopathology results were not recorded in the TB register.

**Table 3. Extrapulmonary manifestation of tuberculosis by sex, age group and place of residence among the cohort of TB patients.**

| | Site-NOS | Pleural | LN-EXT | LN-INT | Abdomen | OAS | OAOS | CNS | Dis/mil | Other | Tot. EPTB | Tot. TB | EPTB% |
|---|---|---|---|---|---|---|---|---|---|---|---|---|---|
| All | 1339 (8.5) | 4668 (29.6) | 3400(21.5) | 181(1.1) | 3313(21.0) | 910(5.8) | 573(3.6) | 725(4.6) | 94(0.6) | 587(3.7) | 15790 | 54092 | 29.2% |
| Median Age (IQR) | 20 (10,35) | 28 (19,45) | 22 (15,32) | 21 (16,32) | 17 (6,30) | 34 (23,50) | 30 (19,48) | 21 (12,40) | 24 (18,40) | 26 (17,38) | 24 (15,39) | 28 (18,46) | |
| **Sex** | | | | | | | | | | | | | |
| Female | 693(8.4) | 2,073 (25.1) | 2,093 (25.3) | 104(1.3) | 1,786 (21.6) | 491(5.9) | 295(3.6) | 365(4.4) | 45(0.5) | 326(3.9) | 8271 | 27,118 | 30.5% |
| Male | 646(8.6) | 2,595 (34.5) | 1,307 (17.4) | 77(1.0) | 1,527 (20.3) | 419(5.6) | 278(3.7) | 360((4.8) | 49(0.7) | 261(3.5) | 7519 | 26,974 | 27.9% |
| FMR | 1.07 | 0.8 | 1.6 | 1.35 | 1.17 | 1.17 | 1.06 | 1.01 | 0.92 | 1.25 | 1.1 | 1.01 | |
| **Age group*** | | | | | | | | | | | | | |
| 0–14 | 437 (12.1) | 504(14.0) | 768(21.3) | 41(1.1) | 1385(38.4) | 74(2.1) | 67(1.9) | 218(6.0) | 12(0.3) | 101(2.8) | 3607 | 8338 | 43.3% |
| 15–24 | 334(7.5) | 1,423 (31.8) | 1,151 (25.7) | 70(1.6) | 795(17.8) | 182(4.1) | 148(3.3) | 182(4.1) | 35(0.8) | 154(3.4) | 4,474 | 14,406 | 31.1% |
| 25–34 | 214(7.4) | 905(31.5) | 681(23.7) | 27(0.9) | 488(17.0) | 202(7.0) | 108(3.8) | 94(3.3) | 11(0.4) | 144(5.0) | 2,874 | 9,348 | 30.7% |
| 35–44 | 129(7.8) | 546(33.1) | 336(20.4) | 18(1.1) | 250(15.2) | 145(8.8) | 75(4.5) | 60(3.6) | 14(0.8) | 76(4.6) | 1,649 | 6,369 | 25.9% |
| 45–54 | 110(8.1) | 493(36.5) | 227(16.8) | 10(0.7) | 180(13.3) | 123(9.1) | 88(6.5) | 58(4.3) | 9(0.7) | 53(3.9) | 1,351 | 6,297 | 21.5% |
| 55–64 | 63(6.6) | 397(41.4) | 128(13.4) | 9(0.9) | 115(12.0) | 98(10.2) | 51(5.3) | 55(5.7) | 4(0.4) | 38(4.0) | 958 | 5,033 | 19.0% |
| 65+ | 52(5.9) | 400(45.8) | 108(12.4) | 6(0.7) | 98(11.2) | 86(9.8) | 36(4.1) | 58(6.6) | 9(1.0) | 21(2.4) | 874 | 4,279 | 20.4% |
| All≥15yrs | 902(7.4) | 4164 (34.2) | 2631(21.6) | 140(1.2) | 1926(15.8) | 836(6.9) | 506(4.2) | 507(4.2) | 82(0.7) | 486(4.0 | 12180 | 45732 | 26.6% |
| **Place of residence** | | | | | | | | | | | | | |
| PJB | 463(8.3) | 2019 (36.2) | 1305 ((23.4) | 86(1.5) | 672(12.0) | 291(5.2) | 288(5.2) | 200(3.6) | 10(0.2) | 246(4.4) | 5580 | 23530 | 23.7% |
| SND | 101(2.9) | 1190 (33.9) | 1002(28.5) | 32(0.9) | 602(17.1) | 272(7.7) | 94(2.7) | 80(2.3) | 21(0.6) | 120(3.4) | 3514 | 15808 | 22.2% |
| KP | 454 (11.6) | 918(23.4) | 635(16.2) | 57(1.5) | 1350(34.4) | 163(4.1) | 101(2.6) | 90(2.3) | 30(0.8) | 132(3.4) | 3930 | 7802 | 50.4% |
| BTN | 204 (25.5) | 83(10.4) | 59(7.4) | 5(0.6) | 124(15.5) | 56(7.0) | 37(4.6) | 205 (25.6) | 12(1.5) | 16(2.0) | 801 | 2111 | 37.9% |
| FATA | 7(1.5) | 104(22.0) | 47(10.0) | (0.0) | 235((49.8) | 24((5.1) | 8(1.7) | 19(4.0) | 1((0.2) | 27(5.7) | 472 | 938 | 50.3% |
| GB | 5(1.1) | 44(10.1) | 105(24.0) | (0.0) | 201(46.0) | 31(7.1) | 25(5.7) | 13(3.0) | 3(0.7) | 10(2.3) | 437 | 1693 | 25.8% |
| AJK | 9(2.6) | 136(39.2) | 65(18.7) | 1(0.3) | 57(16.4) | 26(7.5) | 12(3.5) | 13(3.7) | 5(1.4) | 23(6.6) | 347 | 924 | 37.6% |
| ICT | 96(13.5) | 174(24.5) | 182(25.7) | (0.0) | 72(10.2) | 47((6.6) | 8(1.1) | 105 (14.8) | 12(1.7) | 13(1.8) | 709 | 1286 | 55.1% |

EPTB-Extrapulmonary TB, Site-NOS-EPTB site not specified, LN-EXT-lymphatic extra-thoracic, LN-INT-Lymphatic intrathoracic, OAS-Osteoarticular spine, OAOS-Osteoarticular other than the spine, CNS-Central Nervous system, Dis/Mil–Disseminated /Miliary TB. PJB-Punjab, SND-Sindh, KP-Khyber Pakhtunkhwa, BTN-Balochistan, AJK-Azad Jammu Kashmir, GB-Gilgit Baltistan, FATA-Federally administered tribal areas, ICT-Islamabad Capital territories.

* Age was missing from 22 records including 3 EPTB patients

## Treatment outcomes

Overall treatment success for EPTB cases was higher compared to both bacteriology confirmed and clinically diagnosed PTB. The treatment success was close to 90% for people having pleural, lymphatic and abdominal TB. Treatment success was significantly lower ($P < .001$) for patients with TB meningitis compared to other forms (Table 4).

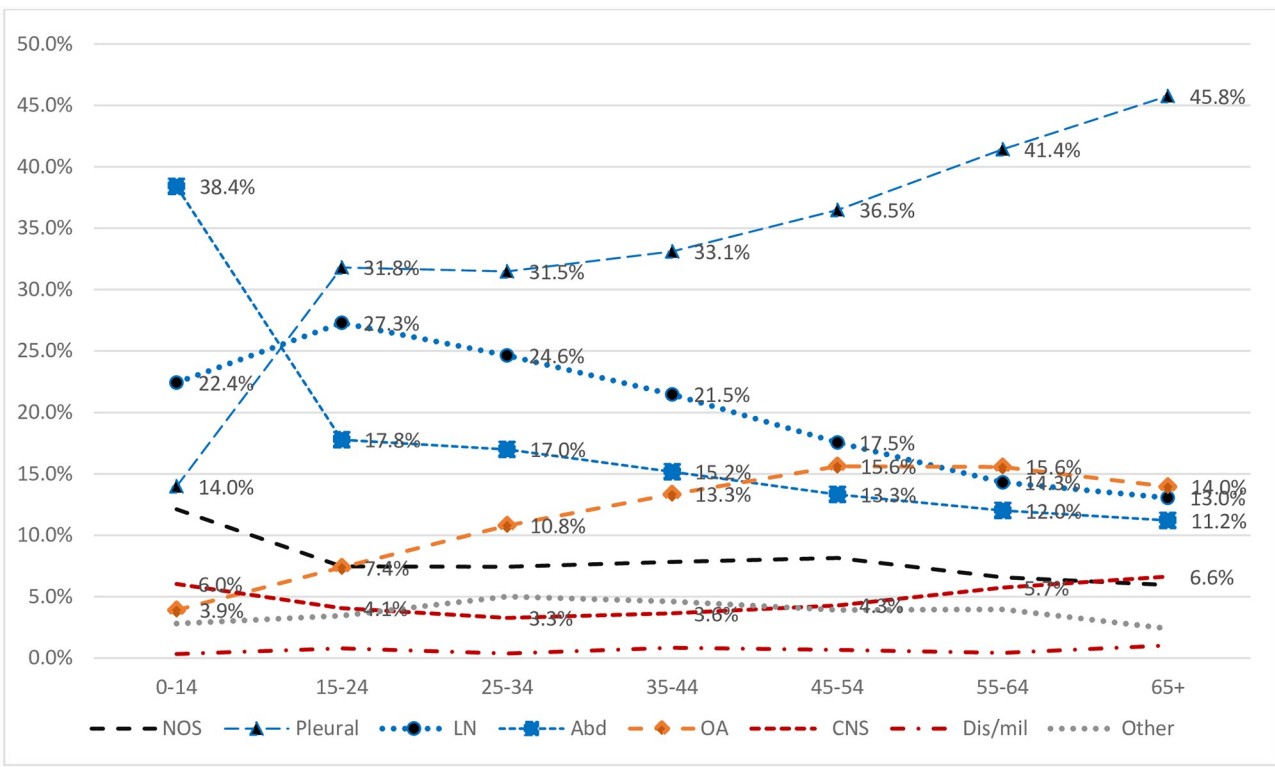

**Fig 2. Frequency of different extrapulmonary manifestations by age group.** NOS: EPTB site not specified, LN-lymphatic, Abd-abdomen; OAS-Osteoarticular, CNS-Central Nervous system, Dis/mil-Disseminated /Miliary.

## Discussion

To our knowledge, this is the first multicenter study where routine TB program data is used to describe demographic characteristics, clinical manifestations and treatment outcomes of EPTB patients in Pakistan. We performed a record review of 54,092 TB patients including 15,790 EPTB cases, notified in 2016 by HCF conveniently selected from all tiers of health care covering all provinces/regions in the country. The study population comprised 21.6% of EPTB and 13.1% of PTB cases notified countrywide in 2016. We purposefully went for oversampling of EPTB to include jurisdictions that appeared to be particularly sensitized to the issue and potentially would provide complete records to study a wider spectrum of EPTB manifestations. There was thus a higher proportion of EPTB in the study sample (29%) compared to national notification (20%) in the same year [1].

The variations in proportion of EPTB among TB cases notified in different provinces/ regions is already observed in routine TB notification data. Proportion of EPTB in countries neighboring Pakistan, varies from 5% in China (HBC), 18% in India (HBC), 25% in Iran and 29% in Afghanistan [2]. Interestingly compared to national average, proportion of EPTB is higher in provinces/regions in the North West (KP, FATA, BTN) neighboring Afghanistan and Iran and a relative lower in proportion in the South East provinces (PJB &SND) neighboring India. In TB population studied, proportion of EPTB although was higher but provincial ranking was consistent with national notification trend. There are some evidences suggesting that PTB incidence decreases as geographical altitude increases [14–18] without effect on EPTB incidence [18]. In Pakistan, provinces and regions in North West with higher proportion of EPTB among notified TB cases, show a trend of lower PTB case notification rate

**Table 4. Treatment outcomes of patients having bacteriology confirmed pulmonary tuberculosis, clinically diagnosed pulmonary tuberculosis and extrapulmonary tuberculosis.**

| | PULMONARY TB | | EXTRA PULMONARY TB | | | | | | | | |
|---|---|---|---|---|---|---|---|---|---|---|---|
| | Bacteriology confirmed | Clinically diagnosed | All | Pleural | Lymphatic | Abdominal | Osteo-Articular | CNS | Diss./ Mil | Other | NOS |
| TB cases(n) | 18872 | 19430 | 15790 | 4668 | 3581 | 3313 | 1483 | 725 | 94 | 587 | 1339 |
| Successfully Treated* | 14959:79.3% (78.7–79.8) | 16803:86.5% (86.0–87.0) | 14125:89.5% (89.0–89.9) | 4233:90.7% (89.8–91.5) | 3217:89.8 (88.8–90.8) | 3018:91.1% (90.1–92.0) | 1288:86.9% (85.0–88.5) | 539:74.3% (71.0–77.5) | 78:83.0% (73.8–89.9) | 496:84.5% (81.3–87.3) | 1256:93.8% (92.4–95.0) |
| Lost to Follow up* | 1147: 6.1% (5.7–6.4) | 844:4.3% (4.1–4.6) | 653:4.1% (3.8–4.5) | 207:4.4% (3.9–5.1) | 135:3.8% (3.2–4.4) | 66:2.0% (01.5–2.5) | 70;4.7% (3.7–5.9) | 110;15.2% (12.6–18.0) | 9:9.6% (4.4–17.4) | 28:4.8% (3.2–6.8) | 28:2.1% (1.4–3.0) |
| Treatment-Failure* | 430;2.3% (2.1–2.5) | 71;0.4% (0.3–0.5) | 33;0.2% (0.1–0.3) | 6;0.1% (0.05–0.3) | 8;0.2% (0.09–0.4) | 14;0.4% (0.2–0.7) | 3;0.2% (0.04–0.6) | 1;0.1% (00.0–0.7) | 0;0.0% (0.0–0.03) | 1;0.2% (0.0–0.9) | 0;0.0% (0.0-.0.3) |
| Died* | 616;3.3% (3.0–3.5) | 410;2.1% (1.9–2.3) | 207;1.3% (1.1–1.5) | 67;1.4% (1.1–1.8) | 33;0.9% (00.6–1.3) | 40;1.2% (0.9–1.6) | 15;1.0% (0.6–1.7) | 32;4.4% (3.0–6.2) | 4;4.3% (1.2–10.5) | 3;0.5% (0.1–1.5) | 13;1.0% (0.5–1.7) |
| Transferred out * | 382;2.0% (1.8–2.2) | 112;0.6% (0.5–0.7) | 78;0.5% (0.4–0.6) | 16;0.3% (0.2–0.6) | 29;0.8% (0.5–1.2) | 5;0.2% (0.05–0.4) | 12;0.8% (0.4–1.4) | 8;1.1% (0.5–2.2) | 0;0.0% (0.0–0.03) | 1;0.2% (0.0–0.9) | 7;0.5% (0.2–1.1) |
| Not evaluated/ recorded* | 1338;7.1% (6.7–7.5) | 1190;6.1% (5.8–6.5) | 694;4.4% (4.1–4.7) | 139;3.0% (2.5–3.5) | 159;4.4% (3.8–5.2) | 170;5.1% (4.4–5.9) | 95;6.4% (5.2–7.8) | 35;4.8% (3.4–6.7) | 3;3.2% (0.7–9.0) | 58;9.9% (7.6–12.6) | 35;2.6% (1.8–3.6) |

*Data shown are number of patients;% (95%CI)

CNS-Central Nervous system, Diss/Mil-Disseminated/Miliary TB, NOS-Site not specified

compared to national average (S2 Table). However, differences in notifications of PTB and EPTB cannot be explained based on this correlation alone and further studies including sub national surveys are needed to determine prevalence of TB, coverage and accessibility to health care services, health seeking behaviors of patients with extrapulmonary manifestations compared to pulmonary symptoms and practices of doctors managing EPTB in different geographical settings.

In total study population, male and female were equal in number but differences were observed in gender distribution both among PTB and EPTB cases notified in different provinces. Similar to the global trends higher number of male among PTB cases and higher odds of female presenting with EPTB was seen in residents of South-Eastern provinces (SND, PJB) and ICT.[1–5,7,8] In contrast, a higher female to male ratio was seen among PTB patients from North-Western provinces (KP, BTN), consistent with trend seen in routine notification [19,20] and similar to the pattern in bordering Afghanistan [2]. However likelihood of both sexes presenting with EPTB manifestations was similar in these regions (BTN, KP, FATA, and AJK). Differences in the relative frequency of EPTB by sex, race, ethnicity, and provenance have been reported by others [3–5, 8, 21–23]. Gender differences in notification rates are known to reflect differences in TB epidemiology, and/or gender differences in access to care [24]. In Pakistan, based on TB prevalence survey findings, it is estimated that males have 1.8 times higher burden of PTB [1,2,25] but gender differences are minimal among notified TB cases and prevalence to notification ratio clearly explains gaps in case detection of PTB in males. However as opposed to PTB, the true burden and gaps in detection of EPTB cannot be determined objectively as EPTB is typically excluded from the measurement of prevalence [2]. Regardless, variation in case notification rates between provinces, similarity was seen in adult case notification pattern, with higher number of females among notified PTB and EPTB

patients in age group of 15-34yr, which was similar to the PTB notifications pattern reported in Europe in middle of last century.[24]. Pakistan has one of the highest levels of child and maternal under nutrition worldwide and large social and geographical inequalities are noted in child and maternal nutrition [26,27]. According to WHO estimates approximately 150,000 incident TB cases in Pakistan are associated with malnutrition [2]. Further studies are needed to establish evidence based linkages between TB in young females to poverty, under nutrition, sex-specific social and biological characteristics.

Consistent with other studies, we reported a strong association between EPTB and children, [3, 4, 8] with EPTB affecting children twice as frequently as adults (OR = 2.0). The most intriguing finding was high proportion of children (n = 1385, 38.4%) having abdominal TB and more than 80% of these children belonged to provinces in North-West (KP, FATA, and GB). Beside children, higher rates of abdominal TB was also reported in adult TB patients from KP and FATA. While surveillance data analysis of EPTB patients from high income countries have reported low prevalence (<5%) of abdominal TB [3,4, 8], but prevalence varying from 6.8% (Afghanistan), 12.8% (India), 14.8% (Nepal), 15.4% (Australia) and 17.5% (Saudi Arabia) are reported mostly in hospital-based small studies [28–32]. The drivers of TB in children including the high levels of chronic and acute malnutrition do still exist in Pakistan, with over 10 million stunted children [27]. The malnutrition alone, however, does not explain the higher prevalence of abdominal TB among children in the North-West regions as a higher prevalence of stunting (> national average of 40.2%) is also reported in SND in the South. There is a need to study high rates of abdominal TB in children and in particular clinical criteria used for diagnosis and prevailing empiric TB treatment practices.

Consistent with trend reported in other countries, the two most common extrapulmonary manifestations reported in our study population were pleural and lymphatic, collectively making 50% of all EPTB cases [3–5, 7, 8, 21, 22, 28–32]. Similarity in frequency of different EPTB manifestations was noted in adult population, with pleural TB being the most common across all provinces/regions. Although lymphatic TB was the second commonest form, but only a very few patients (1%) were reported having intra-thoracic lymphatic TB. Contrary to our findings, high predilection for intra-thoracic lymphatic TB in young ages is reported by others [3, 4]. Likely reasons for under reporting in our settings include lack of training and/or access to radiology facilities or under-recording of findings in TB registers.

We reported osteoarticular TB in 9.3% (1483) of EPTB patients and spinal TB was noted to be more common compared to other forms of bones/joint TB. The median ages of patients with osteoarticular TB was higher compared to all other forms of EPTB. Our findings are consistent with prevalence reported in high income countries [3, 4, 5, 7] but prevalence is lower compared to Benin (25.4%) and Ghana (17.5%) [8, 33].

Among EPTB patients in our study, 6.0% of children and 4.2% adults were reported having CNS TB, frequency was higher compared to high-income countries [3,4,7] but lower than reported by studies in low income and high HIV prevalence countries [8,33]. Among ten tertiary care hospital included in our study, with the exception of three, proportion of CNS TB reported by each was lower than total average. Based on these findings, possibility of reporting a higher prevalence of CNS TB due to selection bias can be argued, nevertheless it does indicates serious gaps in notifications and weak linkages between reporting units and other specialties treating CNS TB in tertiary care settings. Beside weak linkages, other possible reasons for underreporting includes patients with serious forms of TB dying before reaching HCF or dying undiagnosed in hospitals.

We reported a low frequency of "other" group of EPTB manifestations including genitourinary tuberculosis (3.7%) in population studied. Although possibility of underreporting cannot be excluded, but our findings are consistent with reports from high income countries, where a

relative higher predilection of genitourinary tuberculosis is reported in whites compared to non-whites. [3, 5, 23]

EPTB disease site was not recorded in 8.5% of patients included in the study. In TB-burdened countries EPTB is one of the important causes of fever of unknown origin [34]. In TB registers where disease site was recorded for majority of the notified EPTB cases, one plausible explanation for missing details is, that these patients were most likely initiated on empiric TB treatment for fever of unknown origin without further investigations either due to lack of diagnostic facilities or resources.

Of all EPTB cases included in the study, a high majority of the patients were successfully treated consistent with findings from other studies [4, 8]. Proportion of patients with history of previous treatment was low in EPTB (4.0%) compared to PTB (10.2%) cohort, which most likely was one of the reasons for a higher treatment success rate compared to PTB. However treatment outcomes are usually reported, using standard classifications; and "treatment completion" is considered a successful outcome [6, 8]. Possibilities cannot be excluded of reporting treatment completion for patients without clinical improvement particularly in case of undiagnosed drug resistance or disease other than EPTB, as well as for patient with clinical improvement who were initiated on empiric TB treatment for trivial nonspecific infection.

Furthermore, treatment outcomes of different non-pulmonary manifestations of TB cannot be lumped together as if they are a single entity [3]. Treatment success rate were higher for pleural, lymphatic and abdominal TB (90%), but was significantly lower for CNS TB (74.3%, 95%CI 71.0–77.5%). Reported mortality in patients with CNS TB was 4.4% but more than 15% were reported lost to follow up with possibility of underreporting of mortality rate. The poor outcome of CNS TB and other severe forms of TB are masked by the overall high treatment success and are thus ignored.

EPTB can affect any part of the body, and due to the heterogeneity in clinical manifestations, and difficulties in obtaining specimens, the definite diagnosis can be especially challenging. In high-income countries, 50–60% of the notified EPTB cases are bacteriology confirmed [3, 5, 8, 32]. On the contrary, in our study, less than 10% of patient were tested and only 0.5% of the notified EPTB cases were bacteriology confirmed, similar to reports from other low income countries [8]. Besides lack of resources and access to specialized facilities, likely reasons for reliance on clinical diagnosis, includes low sensitivity of widely available AFB microscopy, limited access and complexity associated with TB culture testing and lack of surveillance for EPTB. Possibilities of over and under-diagnosis of EPTB cannot be excluded in these scenarios. EPTB in general is not given high priority on the public health agenda, probably as it does not contribute significantly to the transmission of the disease [4]. With availability of simpler, more sensitive molecular diagnostics [35] there is need for systematic efforts to increase testing of all EPTB specimens for definite diagnosis of TB and drug resistance. This can be achieved by a more decentralized access to new diagnostic tools, multidisciplinary engagements with effective linkages, clear guidance on diagnosis and management of EPTB, capacity enhancement of clinical and laboratory staff and finally comprehensive surveillance system to monitor EPTB notifications in real time.

The main limitations of our study includes use of stratified convenient sampling and voluntary data collection, which makes it quite possible that our findings are not completely representative of the entire country. Secondly, routine clinical records were used with possible multiple problems in the information recorded. Third, almost all of the notified EPTB cases were clinically diagnosed with possibilities of different protocols being used for diagnosis across country.

## Conclusion

The study provides an insight into the demography, prevalent clinical manifestations and treatment outcomes of EPTB. Further studies are needed to explain significant differences observed between provinces and associated specific risk factors. From a public health perspective, there is a need to focus on EPTB accounting 20% of the TB disease burden, and address challenges concerning the quality of diagnosis and treatment of patients presenting with extrapulmonary manifestations of TB.

## Supporting information

**S1 Fig. Pakistan map showing provincial and regional boundaries.**
(PDF)

**S1 Table. Characteristics of the selected health care facilities and TB cases notified.** EPTB–Extra pulmonary tuberculosis, PTB-Pulmonary tuberculosis, HCF-Health care facility, HCP-Health care provider, Pub-public, Pvt–private, THC-Tertiary health care, SHC–Secondary health Care, PHC-Primary health care, GP–General Practitioner, GX- GeneXpert, HP- histopathology, PR- Paper register, ER- electronic file, PJB-Punjab, SND- Sind, KP-Khyber Pakhtunkhwa, BTN-Balochistan, AJK -Azad Jammu Kashmir, GB-Gilgit Baltistan, FATA-Federally administered tribal area, ICT-Islamabad capital Territory.
(PDF)

**S2 Table. National and study sites TB case notifications in 2016 by province and region.**
PTB- Pulmonary tuberculosis, EPTB–Extra pulmonary tuberculosis, N = New TB case,
R = Relapse TB case, GB–Gilgit Baltistan, KPK–Khyber Pakhtunkhwa, AJK-Azad Jamu Kashmir, FATA- Federally administered tribal area, ICT–Islamabad capital Territory, GP- General practioner.
(PDF)

**S3 Table. Age and sex specific pulmonary and extrapulmonary tuberculosis notifications and odds of female (OR) for having extrapulmonary tuberculosis by place of residence.**
F-EPTB = Female Extrapulmonary TB, F-PTB = Female Pulmonary TB cases,
M-EPTB = Male Extrapulmonary TB, M-PTB = Male Pulmonary TB cases.
(PDF)

**S4 Table.** A: Notified tuberculosis cases and extra-pulmonary manifestations of tuberculosis by sex, age groups and health facilities in Punjab, Pakistan during 2016 EPTB–Extra pulmonary tuberculosis, Site-NOS: EPTB site not specified, LN-EXT-lymphatic extra-thoracic, LN-INT-Lymphatic intra thoracic, ABD-Abdomen, OAS–Osteoarticular spine, OAOS-Osteoarticular other than spine,CNS-Central nervous system, DIS/MIL–Disseminated /Miliary TB. B: Notified tuberculosis cases and extrapulmonary manifestation of tuberculosis by sex, age groups and health facilities in Sindh, Pakistan during 2016 EPTB–Extra pulmonary tuberculosis, Site-NOS: EPTB site not specified, LN-EXT-lymphatic extra-thoracic, LN-INT-Lymphatic intra thoracic, ABD-Abdomen, OAS–Osteoarticular spine, OAOS-Osteoarticular other than spine,CNS-Central nervous system, DIS/MIL–Disseminated /Miliary TB. C: Notified tuberculosis cases and extrapulmonary manifestations of tuberculosis by sex, age groups and health facilities in Khyber Pakhtunkhwa, Pakistan during 2016 EPTB–Extra pulmonary tuberculosis, Site-NOS: EPTB site not specified, LN-EXT-lymphatic extra-thoracic, LN-INT-Lymphatic intra thoracic, ABD-Abdomen, OAS–Osteoarticular spine, OAOS-Osteoarticular other than spine,CNS-Central nervous system, DIS/MIL–Disseminated /Miliary TB. D: Notified tuberculosis cases and extrapulmonary manifestations of tuberculosis by sex, age groups and

health facilities in Balochistan, Pakistan, during 2016 EPTB–Extra pulmonary tuberculosis, Site-NOS: EPTB site not specified, LN-EXT-lymphatic extra-thoracic, LN-INT-Lymphatic intra thoracic, ABD-Abdomen, OAS–Osteoarticular spine, OAOS-Osteoarticular other than spine,CNS-Central nervous system, DIS/MIL–Disseminated /Miliary TB. E: Notified tuberculosis cases and extrapulmonary manifestations of tuberculosis by sex, age groups and health facilities in Federally Administered Tribal Areas, Pakistan during 2016 EPTB–Extra pulmonary tuberculosis, Site-NOS: EPTB site not specified, LN-EXT-lymphatic extra-thoracic, LN-INT-Lymphatic intra thoracic, ABD-Abdomen, OAS–Osteoarticular spine, OAOS-Osteoarticular other than spine,CNS-Central nervous system, DIS/MIL–Disseminated /Miliary TB. F: Notified tuberculosis cases and extrapulmonary manifestations of tuberculosis by sex, age groups and health facilities in Gilgit Baltistan, Pakistan during 2016 EPTB–Extra pulmonary tuberculosis, Site-NOS: EPTB site not specified, LN-EXT-lymphatic extra-thoracic, LN-INT-Lymphatic intra thoracic, ABD-Abdomen, OAS–Osteoarticular spine, OAOS-Osteoarticular other than spine,CNS-Central nervous system, DIS/MIL–Disseminated /Miliary TB. G: Notified tuberculosis cases and extrapulmonary manifestations of tuberculosis by sex, age groups and health facilities in Azad Jammu & Kashmir, Pakistan, during 2016 EPTB–Extra pulmonary tuberculosis, Site-NOS: EPTB site not specified, LN-EXT-lymphatic extra-thoracic, LN-INT-Lymphatic intra thoracic, ABD-Abdomen, OAS–Osteoarticular spine, OAOS-Osteoarticular other than spine,CNS-Central nervous system, DIS/MIL–Disseminated /Miliary TB. H: Notified tuberculosis cases and extrapulmonary manifestations of tuberculosis by sex, age groups and health facilities in Islamabad capital Territory, Pakistan during 2016 EPTB–Extra pulmonary tuberculosis, Site-NOS: EPTB site not specified, LN-EXT-lymphatic extra-thoracic, LN-INT-Lymphatic intra thoracic, ABD-Abdomen, OAS–Osteoarticular spine, OAOS-Osteoarticular other than spine,CNS-Central nervous system, DIS/MIL–Disseminated /Miliary TB.
(PDF)

## Acknowledgments

We would like to acknowledge Provincial TB Program managers Punjab, Sindh, Khyber Pakhtunkhwa, Balochistan, FATA, AJK and all health facility staff for facilitation in data collection. We are also thankful to Mr. Safdar Malik (NRL) for data entry.

## Author Contributions

**Conceptualization:** Sabira Tahseen, Faisal Masood Khanzada, Aurangzaib Quadir Baloch, Tehmina Mustafa.

**Data curation:** Sabira Tahseen, Faisal Masood Khanzada.

**Formal analysis:** Sabira Tahseen, Faisal Masood Khanzada.

**Investigation:** Qasim Abbas, Mansoor Manzoor Bhutto, Ahmad Wali Alizai, Shah Zaman, Zahida Qasim, Muhammad Najeeb Durrani, M. Khalid Farough, Atiqa Ambreen, Nauman Safdar.

**Methodology:** Sabira Tahseen, Tehmina Mustafa.

**Project administration:** Sabira Tahseen, Faisal Masood Khanzada.

**Resources:** Aurangzaib Quadir Baloch.

**Software:** Faisal Masood Khanzada.

**Supervision:** Sabira Tahseen, Tehmina Mustafa.

**Validation:** Sabira Tahseen, Faisal Masood Khanzada.

**Writing – original draft:** Sabira Tahseen.

**Writing – review & editing:** Sabira Tahseen, Faisal Masood Khanzada, Aurangzaib Quadir Baloch, Qasim Abbas, Mansoor Manzoor Bhutto, Ahmad Wali Alizai, Shah Zaman, Zahida Qasim, Muhammad Najeeb Durrani, M. Khalid Farough, Atiqa Ambreen, Nauman Safdar, Tehmina Mustafa.

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
