## [Decision Letter · Decision Letter 0]

17 Mar 2020

PONE-D-20-02877

Extrapulmonary tuberculosis in Pakistan- A Nation-wide Multicenter Retrospective study

PLOS ONE

Dear Dr. Tahseen,

Thank you for submitting your manuscript to PLOS ONE. After careful consideration, we feel that it has merit but does not fully meet PLOS ONE’s publication criteria as it currently stands. Therefore, we invite you to submit a revised version of the manuscript that addresses the points raised during the review process.

We would appreciate receiving your revised manuscript by May 01 2020 11:59PM. To enhance the reproducibility of your results, we recommend that if applicable you deposit your laboratory protocols in protocols.io, where a protocol can be assigned its own identifier (DOI) such that it can be cited independently in the future. For instructions see: http://journals.plos.org/plosone/s/submission-guidelines#loc-laboratory-protocols

We look forward to receiving your revised manuscript.

Kind regards,

HASNAIN SEYED EHTESHAM

Academic Editor

PLOS ONE

Additional Editor Comments:

Major Revision

2. We note that Figure S1 in your submission contains map images which may be copyrighted.

a. You may seek permission from the original copyright holder of Figure S1 to publish the content specifically under the CC BY 4.0 license. 

b. If you are unable to obtain permission from the original copyright holder to publish this figure under the CC BY 4.0 license or if the copyright holder’s requirements are incompatible with the CC BY 4.0 license, please either i) remove the figure or ii) supply a replacement figure that complies with the CC BY 4.0 license. Please check copyright information on all replacement figures and update the figure caption with source information. If applicable, please specify in the figure caption text when a figure is similar but not identical to the original image and is therefore for illustrative purposes only.

Reviewers' comments:

Reviewer's Responses to Questions

**Comments to the Author**

1. Is the manuscript technically sound, and do the data support the conclusions?

Reviewer #1: Yes

Reviewer #2: Partly

2. Has the statistical analysis been performed appropriately and rigorously? 

Reviewer #1: Yes

Reviewer #2: I Don't Know

3. Have the authors made all data underlying the findings in their manuscript fully available?

Reviewer #1: Yes

Reviewer #2: Yes

4. Is the manuscript presented in an intelligible fashion and written in standard English?

Reviewer #1: Yes

Reviewer #2: Yes

5. Review Comments to the Author

Reviewer #1: Authors have chosen a relevant topic. Current manuscript is technically sound and relevant to the subject concerned. Study covers large sample size which provides insight into demography and clinical manifestation of extra pulmonary cases.

Reviewer #2: The manuscript by Tahseen et. al, has provided some insight into epidemiology of extrapulmonary tuberculosis (EPTB) in Pakistan. Despite various limitations, authors have tried to study the prevalence of EPTB across various regions, genders and age groups in Pakistan. ‘EPB’ word in the abstract needs to be corrected as ‘EPTB’. It would also be important for the reader to know about the possible reasons behind the association of specific risk factors (sex, race, ethnicity and provenance) with epidemiology of EPTB. Viewpoints of the authors regarding such correlation with appropriate references are lacking in this manuscript. I think the authors should very critically discuss their findings under the discussion part. A thorough discussion will complement the limited data availability in this study. This would also tremendously increase the value of the manuscript.

6. PLOS authors have the option to publish the peer review history of their article (what does this mean?). If published, this will include your full peer review and any attached files.

Reviewer #1: No

Reviewer #2: No

---

## [Author Response · Author response to Decision Letter 0]

5 Apr 2020

We wish to thank you for very thorough and constructive comments on our manuscript entitled, “Extrapulmonary tuberculosis in Pakistan- A Nation-wide Multicenter Retrospective study" (Original Article)

We have revised the manuscript based on your comments. Here we provide point-to-point replies. The changes in the paper are marked in red color in the resubmitted manuscript. 

Comments to the Author

We have now edited the manuscript according to style template given at 

http://www.journals.plos.org/plosone/s/file?id=wjVg/PLOSOne_formatting_sample_main_body.pdf

2. We note that Figure S1 in your submission contains map images which may be copyrighted.

We have now obtained permission from the original copyright holder of Figure S1 to publish the content specifically under the CC BY 4.0 license. Source and caption of the figure is added with the figure

Content permission on PLOSONE recommended format. Three file are uploaded named as “Other1” approval on PLOSONE recommended format, “Other2” email communication and “Other3” terms and condition of D-maps.com

Thank you for your guidance we have now edited authors’ affiliation as recommended in 

http://www.journals.plos.org/plosone/s/file?id=ba62/PLOSOne_formatting_sample_title_authors_affiliations.pdf

Review Comments to the Author

Reviewer #1: Authors have chosen a relevant topic. Current manuscript is technically sound and relevant to the subject concerned. Study covers large sample size which provides insight into demography and clinical manifestation of extra pulmonary cases.

Thank you very much 

Reviewer #2: The manuscript by Tahseen et. al, has provided some insight into epidemiology of extrapulmonary tuberculosis (EPTB) in Pakistan. Despite various limitations, authors have tried to study the prevalence of EPTB across various regions, genders and age groups in Pakistan. ‘EPB’ word in the abstract needs to be corrected as ‘EPTB’. It would also be important for the reader to know about the possible reasons behind the association of specific risk factors (sex, race, ethnicity and provenance) with epidemiology of EPTB. Viewpoints of the authors regarding such correlation with appropriate references are lacking in this manuscript. I think the authors should very critically discuss their findings under the discussion part. A thorough discussion will complement the limited data availability in this study. This would also tremendously increase the value of the manuscript.

Thank you very much for your comment, we have now edited the discussion section and critically discussed the key findings and added our view points. 

Changes in manuscript; 

• All changes are marked in red color in the resubmitted manuscript. 

• Some text is rearranged in results section to better align with edits in discussion without changing the content.

• Colored figures (Figure 1 and 2) uploaded to replace black and white, uploaded originally with manuscript for more clarity without any change in the content 

• Support table: Table S4A-H: In all tables an additional row is added under age group -“Adults (All)” for cumulative number for adult of all age groups .

---

## [Editor Report · Decision Letter 1]

8 Apr 2020

Extrapulmonary tuberculosis in Pakistan- A nation-wide multicenter retrospective study

PONE-D-20-02877R1

Dear Dr. Tahseen,

We are pleased to inform you that your manuscript has been judged scientifically suitable for publication and will be formally accepted for publication once it complies with all outstanding technical requirements.

With kind regards,

HASNAIN SEYED EHTESHAM

Academic Editor

PLOS ONE

Additional Editor Comments (optional):

The Authors have revised the manuscript. There was the issue of copyright which has now been addressed. Other comments have been taken care in the revised manuscript.
---

## [Editor Report · Acceptance letter]

13 Apr 2020

PONE-D-20-02877R1 

Extrapulmonary tuberculosis in Pakistan- A nation-wide multicenter retrospective study 

Dear Dr. Tahseen:

I am pleased to inform you that your manuscript has been deemed suitable for publication in PLOS ONE. Congratulations! Your manuscript is now with our production department. 

With kind regards,

on behalf of

Prof HASNAIN SEYED EHTESHAM 

Academic Editor

PLOS ONE